# Towards an Ergonomic Assessment Framework for Industrial Assembly Workstations—A Case Study

**Ana Colim** [1],*, **Carlos Faria** [1], **Ana Cristina Braga** [2], **Nuno Sousa** [2], **Luís Rocha** [1], **Paula Carneiro** [2], **Nélson Costa** [2] and **Pedro Arezes** [2]

[1]  DTx Colab, 4800-058 Guimarães, Portugal; carlos.faria@dtx-colab.pt (C.F.); luis.rocha@dtx-colab.pt (L.R.)
[2]  ALGORITMI Centre, University of Minho, 4800-058 Guimarães, Portugal; acb@dps.uminho.pt (A.C.B.); nuno.sousa@dps.uminho.pt (N.S.); pcarneiro@dps.uminho.pt (P.C.); ncosta@dps.uminho.pt (N.C.); parezes@dps.uminho.pt (P.A.)
*   Correspondence: ana.colim@dtx-colab.pt

**Abstract:** Work-related musculoskeletal disorders (WMSD) are one of the main occupational health problems. The best strategy to prevent them lies on ergonomic interventions. The variety of industrial processes and environments, however, makes it difficult to define an all-purpose framework to guide these ergonomic interventions. This undefinition is exacerbated by recurrent introduction of new technologies, e.g., collaborative robots. In this paper, we propose a framework to guide ergonomics and human factors practitioners through all stages of assessment and redesign of workstations. This framework was applied in a case study at an assembly workstation of a large furniture enterprise. Direct observation of work activity and questionnaires were applied to characterize the workstations, the process, and the workers' profiles and perceptions. An ergonomic multi-method approach, based on well-known and validated methods (such as the Finnish Institute of Occupational Health and Rapid Upper Limb Assessment), was applied to identify the most critical risk factors. We concluded that this approach supports the process redesign and tasks' allocation of the future workstation. From these conclusions, we distill a list of requirements for the creation of a collaborative robot cell, specifying which tasks are performed by whom, as well as the scheduling of the human-robot collaboration (HRC).

**Keywords:** ergonomics and human factors; risk assessment; WMSD prevention; design of assembly workstations for cobots

## 1. Introduction

Work-related musculoskeletal disorders (WMSD) are one of the main occupational health problems in the European Union and a major cause of occupational absenteeism and decreased productivity [1,2]. The incidence of WMSD is higher than normal in industrial furniture manufacturing due to the characteristics of the involved tasks [2].

The development of WMSD is mainly attributed to three factors, (i) occupational risk, (ii) individual characteristics, and (iii) social factors [3]. Occupational risk factors include awkward postures, repetitive tasks, frequent and/or excessive tasks involving the handling of heavy loads, and thermal discomfort. The individual characteristics are related to individual limitations or health problems. Finally, social factors such as family and economic problems may interfere with motivation and attention during work. According to the European Agency for Safety and Health at Work, organizational and psychosocial factors such as high demand for work or low autonomy, and low job satisfaction can potentiate the WMSD risk [4].

On top of the detrimental health impact of WMSD, these disorders are also financially harmful to the economy in general, being urgent its prevention and mitigation [5,6]. Scientific literature shows that ergonomic intervention is the best strategy to prevent WMSD [7]. The ergonomic intervention aims to redesign the workstation and process to improve health, safety, and productivity [8].

Contrary to other occupational diseases that result from exposure to specific hazards, most WMSD have a multifactorial origin. In the prevention domain, several methods were presented and validated for WMSD risk assessment. These methods fall into the following categories: (i) self-reports and checklists; (ii) observational methods; (iii) direct measurements [9,10].

The self-reports and checklists include the tools more generic according to their field of application (i.e., range of working tasks). At this level, previously validated questionnaires, such as the Nordic Musculoskeletal Questionnaire (NMQ) [11,12], are conducted to collect the workers' perceptions. Checklists, such as the checklist for "Ergonomic Hazard Identification" of National Institute for Occupational Safety and Health [13], are also applied to identify risk factors.

The observational methods rely on the direct observation of the work activity to conduct the risk assessment. These methods consider risk factors such as task frequency, task duration, and load handling and assess the impact of these risks based on the external physical workload. Examples of validated observational methods for assembly tasks are the Rapid Upper Limb Assessment (RULA) [14] and the Key Indicator Method for assessing physical workload during Manual Handling Operations (KIM-MHO) [15]. The direct measurements ergonomic methods rely on the sensorization of the workers to directly measure the risk factors' effect on physiological and biomechanical parameters. Examples of these sensors are the surface electromyography (EMG) to assess muscular activity and the electronic goniometers to record the range of joint motion [9].

Assembly workers are exposed to a significant physical workload [16]. The repetitiveness of manual tasks is an important risk factor for upper limbs WMSD (e.g., carpal tunnel syndrome, wrist tendonitis, and lateral epicondylitis) [10]. However, the back is the body region with the most serious work-related health problems indicated among European workers, being a problem transversal to a wide variety of jobs [4]. Human-robot collaborative manufacturing has been proposed as a potential solution to improve workplace conditions. This theory was tested in different studies [17–20] with promising results in the reduction of the workload and the decrease of the WMSD risk. The industrial collaborative robots, also known as COBOTS, reduce ergonomic concerns that arise from on-the-job physical and cognitive stress, and further improve on workplace safety, quality, and productivity [18]. Some of the most highly cited reviews on COBOT technology [19–21] defend that with human-robot interaction it is possible to combine the adaptability and fast decision making from humans with the precision and consistency of the robot system.

Human-robot collaboration (HRC) is an appealing prospect to the industry in general: to Small and Medium-sized enterprises due to the high degree of adaptability and flexibility, as well as to mass production companies which are rapidly shifting into mass customization. With ever-changing customer demands at an increasingly faster pace, companies look to proactive answers to boost their competitiveness and take the factories to the next level of automation and manufacturing advancement [19,21].

Smart factories are one of the pillars of the 4th industrial revolution (Industry 4.0). Of those, flexible robotic solutions with intuitive and natural human-machine interfaces and capable of intelligent decision making—COBOTs—are key players. COBOTs or collaborative robots are a sub-type of robots specially tailored to work in close proximity to humans or other robots. Through a closer interaction between the machine and the operator, it enables collaborative scenarios where the continuous accuracy, speed, and repeatability typical of robots can be combined with the innate adaptability, dexterity, perception, and intelligence distinctive of humans. This mutualist relation between both parts leads to a powerful collaborative framework to positively impact productivity, flexibility, and most importantly with a positive net effect on the creation of new jobs rather than replacing workers [18,20,22].

Therefore, the current study is integrated into the first phase of a research project of the Collaborative Laboratory DTx—"*Associação Laboratório Colaborativo em Transformação Digital*".

This laboratory is a non-profit private association, which carries out its activity doing applied research in different areas linked to the digital transformation of industry and society, promoting cooperation between the academic research and the industry. In this case, the project results from a cooperation between the Laboratory DTx and a large Portuguese site of furniture manufacturing, foreseeing the future implementation of collaborative robotics in the manual assembly section. In this industrial context, the company's reports pointed out the need for an ergonomic intervention across the manual assembly workstations, where most of the workers were continuously exposed to WMSD risk factors and presenting already several musculoskeletal complaints. The current study aims to identify the characteristics of the workstation problems and to define the ergonomics requirements as we look forward to integrating a COBOT system in the workflow.

## 2. Materials and Methods

According to the company's medical information, a peak on WMSD was registered with frame assembly workers. During the preliminary observation of the work activity, different WMSD risk factors were identified, such as repetitive manual tasks and the adoption of awkward postures, which supported the ergonomic intervention. This ergonomic intervention was subdivided into the following phases: (i) preliminary ergonomic assessment of the existing assembly workstations; (ii) redesign the process and design of a new assembly process and workstation; (iii) ergonomic assessment of the new workstation; (iv) definition of ergonomic requirements to improve the new workstation with collaborative robotics.

Workers participated in the study voluntarily. All participants signed an Informed Consent Term in agreement with the Committee of Ethics for Research in Social and Humans Sciences of University of Minho (approval number CEICSH 095/2019), and in agreement with the Declaration of Helsinki.

### 2.1. Description of the Initial Problem

The assembly section is composed of 14 similar workstations (Figure 1) with 60 females workers. The frame assembly workstation produces semi-products composed of MDF (medium-density fibreboard) stripes and blocks that are hot-glued in a rigorous and specific structure. From video-recording observations, the work activity was subdivided into the following manual tasks: (i) reach stripes from a supply cart (frequently from above the height of the workers' shoulders); (ii) place MDF stripes (above the eye-level height of the workers); (iii) reach blocks from the box (lower than the work plane, leading to neck and trunk flexion—Figure 1); (iv) apply glue to the blocks (with a glue gun activated by finger pressure); (v) glue blocks on top of the stripes (with arms elevated and wrists twisted); (vi) glue blocks below the stripes; (vii) transfer a set of 3 assembled frames to the pallet (with trunk flexion and torsion).

With the involvement of the company's practitioners, the assembly process was redesigned and divided into two phases: the preassembly and the final frame assembly. A new preassembly workstation was designed for workers with musculoskeletal problems/complaints and/or medical restrictions to perform certain tasks. The preassembly process consists of gluing blocks in pre-determined positions on MDF stripes. The sub-product of the preassembly workstation will later be transformed in the "final assembly" into frames.

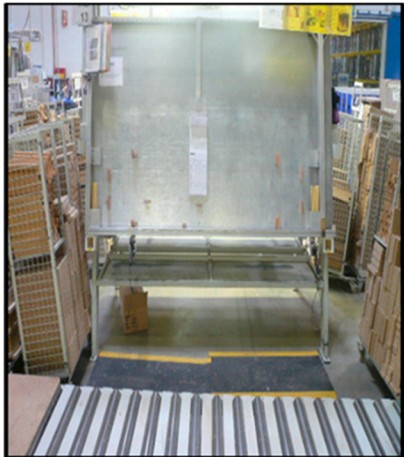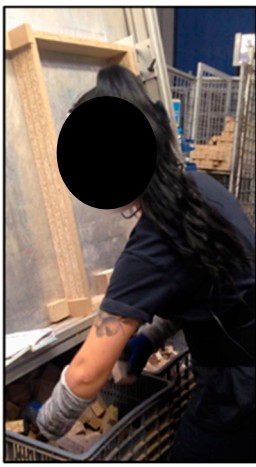

**Figure 1.** Example of the existing assembly workstation.

## 2.2. Existing Assembly Workstations—Ergonomic Assessment

For the ergonomic assessment of the existing workstations, the Ergonomic Workplace Analysis (EWA) method [23] was applied. This method was developed by the Finnish Institute of Occupational Health (FIOH) (henceforth named FIOH method). The FIOH method is the best-known method for the systematic and comprehensive assessment of a workstation. This is an observational method [9], which includes both workers and observers (ergonomic experts) assessments across 14 topics, namely: (1) workspace; (2) general physical activity; (3) lifting tasks; (4) work postures and movements; (5) risk of accident; (6) work content; (7) restrictiveness; (8) workers' communication; (9) decision-making; (10) work repetitiveness; (11) level of required attention; (12) lighting; (13) thermal conditions; and (14) noise. The experts' assessment varies in a 4 or 5 level scale. A score of five (or four) represents a maximum level of risk for the workers on the topic under evaluation. With this method, the workers assess the same 14 topics of the workstation using a four-level rating scale: very poor (- -); poor (-); good (+); very good (+ +). However, to achieve a comprehensive comparison between the experts' and workers' assessments, the workers' rating scale was converted into a numerical scale with four-level (1 to 4 points).

This research phase was also based on the psychophysical approach. This approach has been extensively used in previous studies focusing the WMSD prevention in occupational contexts with handling tasks [24–26]. Globally, these studies demonstrated that, for workstations with handling tasks, ergonomic interventions shown comparable results when based on psychophysical criteria or biomechanical data. These proofs contradict the traditional perception that biomechanical, physiological and psychophysical approaches produce different and even contradictory results. It also supports the application of psychophysical data as an important tool in the definition of acceptable limits of strength and load, or as an indicator of effort perception in preventing physical overload in occupational tasks [27].

For this purpose, a questionnaire was developed in order to collect the psychophysical assessment of the assembly workers. This questionnaire is divided into three categories, each with its own parameters, objectives, and tools applied (Table 1). The categories are defined according to the content assessed by the workers' opinions. These perceptions are assessed trough different scales validated by previous authors.

**Table 1.** Summary of the questionnaire's structure—assembly workers.

| | Questions' Category | Parameters Assessed | Objectives | Tools Applied |
|---|---|---|---|---|
| A. | Workers' characterization | Gender, age, work experience; preferred hand; WMSD. | To characterize the workers' sample with demographic data. | Direct questions. |
| B. | Assessment of Musculoskeletal symptoms | Musculoskeletal symptomology self-reported for the entire body. | To characterize musculoskeletal symptomology of the preassembly workers. | NMQ with closed answers "yes/no" and VAS to assess pain intensity. |
| C. | Assessment of risk factors | 14 topics that influence the ergonomic conditions. | To assess globally the workstation by the workers' perceptions. | Scale based on the FIOH method. |

A pilot application of the questionnaire was done with a group of two workers (randomly selected), after that, few changes were introduced and the questionnaire was applied to the assembly workers. All workers were interviewed during their workday, performing a normal working activity. While the workers had a copy of the questionnaire, the ergonomics expert asked the questions in the form of an interview, noting the worker's answers and providing explanations whenever necessary.

The first part of the questionnaire (A category) is related to the collection of demographic data, such as gender, age, work experience, dominant hand, and previous WMSD, allowing the sample characterization.

The second category was composed of the Portuguese version of the Nordic Musculoskeletal Questionnaire (NMQ) [28]. The NMQ is a standardized questionnaire used to evaluate and to characterize musculoskeletal symptomatology perceived by workers, considering the entire body of them [12]. The NMQ contains a section for the identification of self-reported musculoskeletal symptomatology across nine body regions (neck, shoulders, upper back, lower back, elbows, wrists/hands, hips/thighs, knees, ankles/feet). For each of the referred body regions there was four questions allowing the characterization of the associated symptomatology, namely: (i) related to the last 12 months; (ii) related to the last 7 days; (iii) related to the impediment to perform daily life activities due to any musculoskeletal problem in the last 12 months; (iv) related to the intensity of the felt discomfort/pain. The pain intensity perceived was assessed using a numerical scale (Visual Analogical Scale—VAS), ranging from 0 (without pain) to 10 points (maximum pain).

In order to register workers' perceptions about the assembly workstation, the questions of category C were based on the FIOH method.

The scoring scales mentioned above are based on a closed-form response system. The workers are required to make a choice between certain given options. If these options do not match their opinions, they can freely express it as open comments and/or suggestions for improvements).

*2.3. Preassembly Workstation—Ergonomic Assessment*

As mentioned above, the new workstation (the preassembly workstation) was designed to accommodate the workers with musculoskeletal complaints. A group of 8 workers was selected and reallocated to this workstation. The analysis of this redesign began with the quantification of task times associated with the preassembly process. These times will provide a future reference to measure COBOT workstation efficiency. To uphold a rigorous and precise standard, various cycles of the preassembly process were recorded in video format as part of a time-motion study. The process was initially segmented into a set of well-defined tasks coordinated by the two operators in the work cell.

With the intent of collecting demographic and psychophysical assessments, a questionnaire was also applied to the preassembly workers as well. In this case, the questionnaire was adapted to compare tasks and workstations and thus it has four categories, see Table 2.

**Table 2.** Summary of the questionnaire's structure—preassembly.

| | Questions' Category | Parameters Assessed | Objectives | Tools Applied |
|---|---|---|---|---|
| A. | Workers' characterization | Gender, age, work experience; preferred hand; WMSD. | To characterize the workers' sample with demographic data. | Direct questions. |
| B. | Assessment of Musculoskeletal symptoms | Musculoskeletal symptomology self-reported for the entire body. | To characterize musculoskeletal symptomology of the preassembly workers. | NMQ with closed answers "yes/no" and VAS to assess pain intensity. |
| C. | Perceived exertion associated with the tasks | Perceived exertion for each preassembly task. | To assess physical exertion perceived by the workers; To identify the most demanding tasks. | CR-10 Borg scale. |
| D. | Global assessment of the workstation | Global opinion about the preassembly workstation. | To compare the preassembly with the assembly workstation; To assess workers' opinions about possible improvements to introduce in the preassembly workstation; | 5-Likert scale. |

The first part of this questionnaire has the same categories as the previous one, related to the workers' characterization and assessment of musculoskeletal symptoms. The questionnaire category C related to the self-reported physical exertion at the new workstation was evaluated using the "Category Ratio-10" (CR-10) [29]. Borg [29] argues that the application of scales similar to CR-10 is necessary to quantify and dismiss subjective sensations of physical overload, such as the perception of effort and discomfort. An advantage of the CR-10 scale is that each score is associated with an effort that is perceived by different individuals. This puts in perspective the physical effort perceived by different workers and/or the effect of different work conditions. The last category (D category) was composed of six statements related to the changes introduced at the assembly activity. The workers have to classify the statement using a 5-Likert scale. Workers are requested to indicate their level of agreement with each particular statement. The 5-Likert scale is labeled as follows: 1 = Strongly disagree; 2 = Disagree; 3 = Neutral; 4 = Agree; 5 = Strongly agree.

Additionally, considering the ergonomic challenge associated with this case study, the ergonomic assessment included the application of methods more specific for the work tasks understudy. This more focused assessment allows the definition of ergonomic requirements to improve this workstation. Therefore, considering observation of different work cycles, a total of 38 postures were assessed by the following methods: RULA [14] and KIM-MHO [29]. For the postures selection, the following criteria were applied: (i) representativeness considering the postures more frequent; (ii) selection of the hand (right or left) according to the higher exertion involved; (iii) for tasks with more complex manual work, both hands were separately assessed. The tasks studied were cyclic and exposed workers to a variety of upper extremity activities of varying force, postures and repetitive motion. The task cycle times were measured during the time-motion study.

Relatively to the selection of assessment methods, the RULA and KIM-MHO are observational methods [9] available for assessing WMSD risk during MHO. In order to summarize the main characteristics of the assessment methods applied, Table 3 was constructed taking into account the following parameters: focus' method, risk factors included and not included, output, advantages, and limitations related to the method application. Since this assessment was developed across the different assembly tasks, the need for a comparison between the results obtained by both methods was identified. Based on this need, four global risk levels were defined, integrating/combining the different risk levels considered by each method. These global risk levels are presented in Table 4.

**Table 3.** Comparison of the main characteristics of the methods applied during the ergonomic assessment of the preassembly workstation.

| Method | Focus | Risk Factors Included | Output | Advantages | Limitations |
|---|---|---|---|---|---|
| RULA | Manufacturing and handling tasks in standing or seating posture, as well as work office, allowing the WMSD risk assessment for different body parts (upper limbs, shoulders, neck, and trunk). | Posture of arm, forearm, wrist, neck, and trunk; Repetition/frequency; Load/force. | RULA risk rating with four action levels, indicating the requirements for action on the workplace/task. | Detailed posture analysis considering different body parts; Relevant set of WMSD risk factors is considered. | Time-consuming; Difficulties to assess hand/wrist posture through observation; Repetition/frequency information is vague. |
| KIM-MHO | MHO with repetitive motion and predominantly lower force expenses of the upper extremities. | Duration of tasks; Type of force exertion in the finger-hand area; Repetition of movements/duration of holding; Force transfer/gripping conditions; Hand/arm position and movement; Work organization; Working conditions; Global body posture. | KIM final score with four risk ranges, indicating the possibility of physical overload occurrence and, consequently, the need for workplace redesign. | Complete analysis of the main risk factors for Work-related Upper Limb Disorders (WULD); Application facilitated by illustrations and descriptions for the rating points for the different risk factors. | Assessment of the level of force and posture is less accurate. |

**Table 4.** Global risk levels' definition based on the outputs of the methods considered.

| Global Risk Level | RULA | | KIM-MHO | |
|---|---|---|---|---|
| | Final Score | Meaning | Final Score | Meaning |
| I | 1 or 2 | It indicates that posture is acceptable if it is not maintained or repeated for long periods. | <10 | Low load situation, the health risk from physical overload is unlikely to appear. |
| II | 3 or 4 | It indicates that further investigation is needed and changes may be required. | 10 to <25 | Moderate load situation, physical overload is possible for less resilient persons. For this group redesign of the workplace is helpful. |
| III | 5 or 6 | It indicates that investigation and changes are required soon. | 25 to <50 | Increased load situation, physical overload also possible for normally resilient persons. The redesign of the workplace should be reviewed. |
| IV | 7 | It indicates that investigation and changes are required immediately. | ≥50 | High load situation, physical overload is likely to appear. Workplace redesign is necessary. |

## 2.4. Statistical Analysis

The software IBM®SPSS®Statistics, version 26.0, was applied to analyze the results. A descriptive and exploratory analysis of the data was conducted to calculate the mean values (and standard deviation, SD) of the quantitative variables obtained across the study. Given the nature of the variables involved in the study and the sample size, it was decided to use non-parametric tests.

The workers' assessment in the FIOH method and the prevalence of the musculoskeletal symptoms were expressed in a relative percentage, evidencing the values' distribution. The McNemar test—a specific test of the Chi-square for paired samples—was used to test the concordance between the musculoskeletal pain prevalence between the two periods considered in the NMQ (last 12 months and last 7 days). In addition, considering the answers about FIOH topics, the Friedman test was applied to differentiate groups of topics, since these are dependent variables measured by an ordinal scale. With this test, it was intended to segregate FIOH topics with different workers' answers (as topics with assessments' most positive and negative). The decision rule consists of detecting statistically significant evidence for a $p$-value of less than 0.05.

## 3. Results and Discussion

### 3.1. Characterization of the Initial Problem—Assembly Workers and Workstations

The workforce of the assembly section is composed of 60 female workers divided across 3 work shifts. However, in the current study, a sample of 44 workers was constituted (who volunteered to participate). They are women with an age of 41.4 (± 10.3) years old, with a length of employment of 9.6 (± 2.2) years and all of them are right-handed. Of those, 17 workers had one or more medically diagnosed musculoskeletal problems. The musculoskeletal disorders identified and the number of workers affected are the following: (i) tendinitis at the upper limbs ($n = 9$); (ii) disc herniation ($n = 7$); (iii) carpal tunnel syndrome ($n = 4$); (iv) rheumatoid arthritis ($n = 1$); (v) arthrosis at the back ($n = 1$).

Relatively to the NMQ results (Figure 2), the McNemar test proved the existence of a perfect concordance ($p = 1.000$) between the workers' perceptions for the last 12 months and the last 7 days across the body regions considered. Therefore, the NMQ results presented in Figure 2 are related to the prevalence of musculoskeletal discomfort/pain along the last 12 months. The results demonstrate that all of the inquired workers reported any musculoskeletal symptoms, even the workers without previously disorders diagnosed.

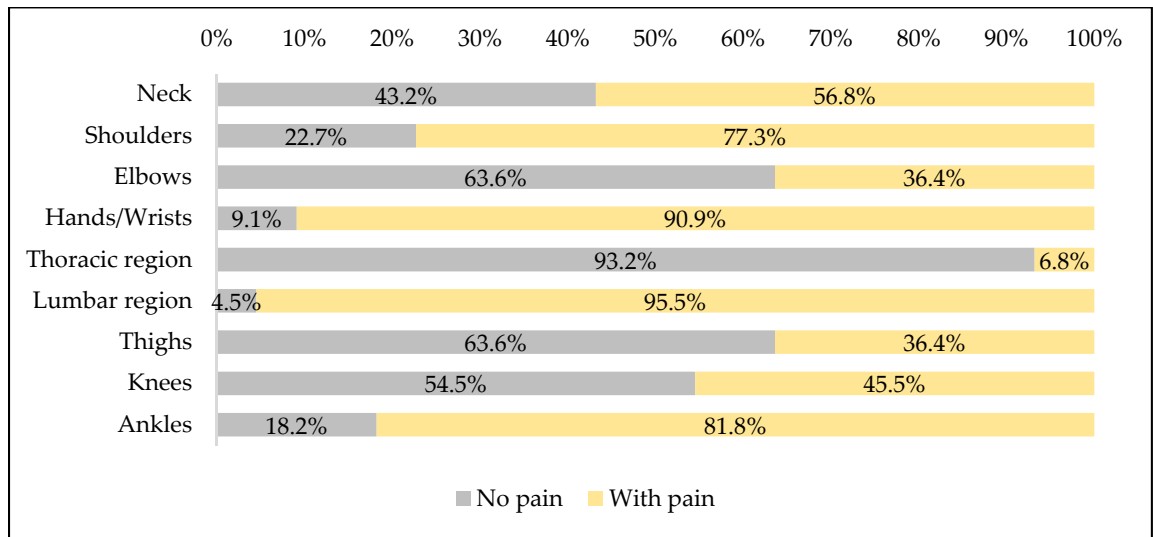

**Figure 2.** Nordic Musculoskeletal Questionnaire (NMQ) results—assembly workers (*n* = 44).

The results obtained by the FIOH method are presented in Table 5, indicating that the workstation presents several constraints for the workers' postures (topic 4), such as arms elevation above the shoulders' height. The work restrictiveness (topic 7) and repetitiveness (topic 10) are two of the parameters with a higher risk (4 points in the FIOH scale) due to the limited work content and short cycle durations (less than 5 min).

The assembly workers (*n* = 44) reported their perceptions through the FIOH method (Figure 3). The Friedman test proves that the answers' distribution is different across the 14 questions, with a test value Q (13) = 320,433. Therefore, the Friedman pairwise comparison reveals that 3 groups of the FIOH topics obtain an equivalent answer distribution. This statistical analysis demonstrates the division of the FIOH topics across the following groups: (i) more positive topics (7; 8; 9; 11; 12); (ii) intermediate (1; 3; 5; 14); and (iii) more negative topics (2; 4; 10; 13). One can also conclude from the analysis that the occupational conditions that require urgent intervention are: physical activity, postures and movements, work repetitiveness and thermal conditions.

Results show that the assembly workers were exposed to significant WMSD risk factors, according to the FIOH assessment and the high percentages of musculoskeletal symptoms prevalence. The findings show that workers reported musculoskeletal pain/discomfort in different body regions. Simultaneously, they assessed their workstations more negatively with respect to certain aspects related to ergonomic risk factors as measured by the FIOH method. Chiasson et al. [30] also found a correlation between pain perceived and the results of FIOH method.

**Table 5.** Finnish Institute of Occupational Health (FIOH) results for the existing assembly workstation—expert assessment.

| | Topic Assessed | Expert Assessment | Comment |
|---|---|---|---|
| 1. | Workspace | 4 | There are serious deviations from the recommendations. Workplace arrangement obliges workers to adopt difficult postures and movements (such as arms elevation above the shoulders). |
| 2. | General physical activity | 3 | The work activity depends on the production method and work organization. For some situations/references production, exist the risk of physical overload due to work peaks. |
| 3. | Lifting tasks | 4 | Heavier loads equal to 14 kg, frequently handling above the shoulders and below the knees. |
| 4. | Work postures and movements | 5 | Need for quick arm movements. |
| 5. | Risk of accident | 3 | Burns for hot glue is very frequent, but the severity is low. |
| 6. | Work content | 3 | The workers perform only a part of the work entity. |
| 7. | Restrictiveness | 4 | Production management requirements restrict method and work pace. |
| 8. | Workers' communication | 2 | The communication between workers is possible, but it is difficulted by the noise. |
| 9. | Decision-making | 2 | The work consists of simple tasks. |
| 10. | Work repetitiveness | 4 | Cycle duration of less than 5 min. |
| 11. | Level of required attention | 3 | Medium level of attention (assembly work) in more than half of the cycle. |
| 12. | Lighting | 2 | Considering the workstation with a lower illuminance level (412 lux) and the recommended value (500 lux). |
| 13. | Thermal conditions | 4 | Air temperature is high for the metabolism associated with the tasks. |
| 14. | Noise | 5 | Noise level above 80 dB(A) and the workers need to communicate (teamwork). |

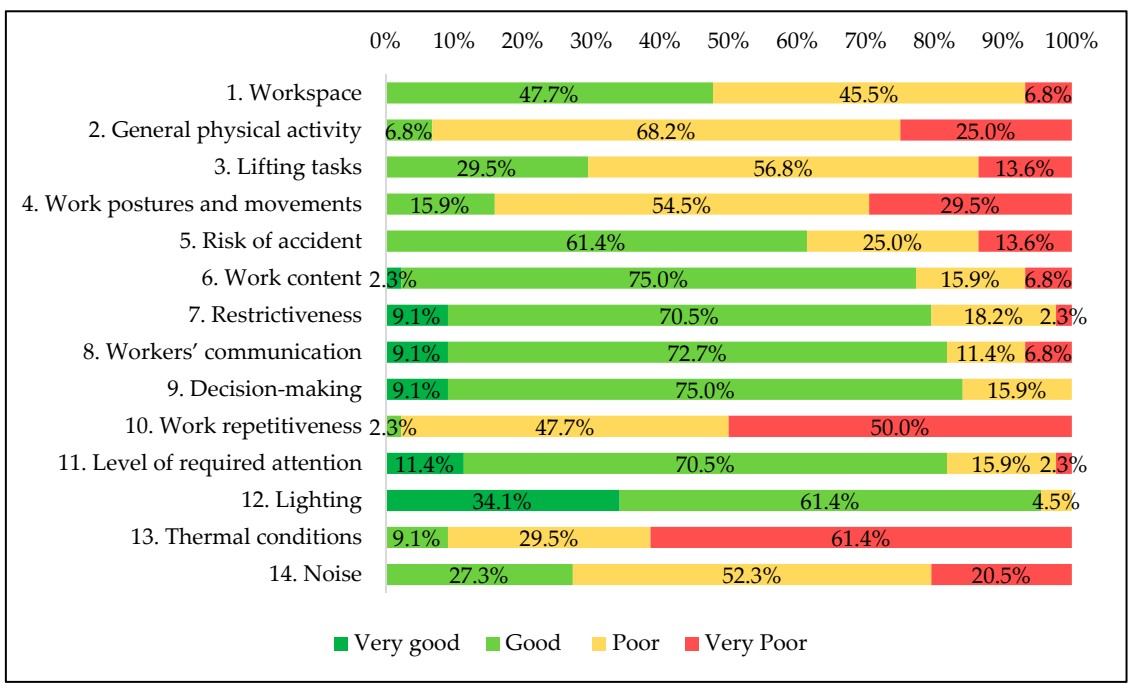

**Figure 3.** FIOH results for the existing assembly workstation perceptions of the workers (*n* = 44).

## 3.2. Redesign of the Assembly Process and Preassembly Workstation

According to the conclusions drawn from the previously collected data, the company took preliminary action to split the frame assembly workstation into two processes, the preassembly and the final assembly of the frames. The principal reason for this separation was the physical limitations of several workers. With the redesign of the process, a new workstation for the preassembly was introduced (Figure 4). Table 6 summarizes the elements of the preassembly work cycle and presents its time mean values (obtained by observation of eight work cycles).

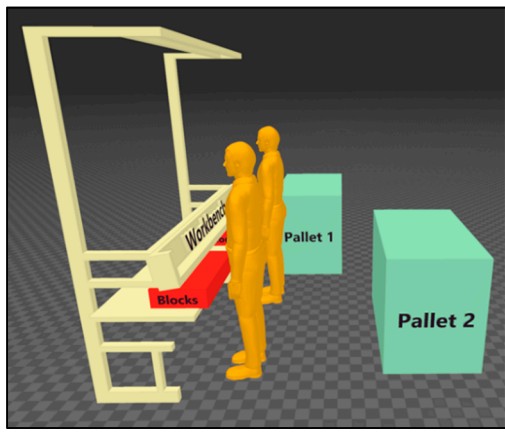

**Figure 4.** Preassembly workstation 3D view.

**Table 6.** List of preassembly tasks and mean times (± Standard Deviation—SD), in seconds (s).

| Tasks | Description | Mean time per cycle (SD) (s) |
|---|---|---|
| T1 | Reach for the stripes from pallet 1 and place them in the assembly workbench (by one worker). | 2.5 ± 0.9 |
| T2 | Pick the blocks from the box. | 7.7 ± 1.2 |
| T3 | Reach for the glue gun. | 2.7 ± 0.3 |
| T4 | Apply glue to the blocks. | 4.1 ± 0.5 |
| T5 | Put down the glue gun. | 3.2 ± 0.6 |
| T6 | Glue the blocks to the stripes. | 9.4 ± 1.9 |
| T7 | Dislodge, rotate and place back the stripe. | 3.1 ± 0.9 |
| T8 | Dislodge the stripes. | 1.6 ± 1.0 |
| T9 | Transfer the stripes to the pallet 2. | 3.2 ± 2.0 |
| T10 | Resupply the glue gun. | 0.7 ± 0.0 |

The design of this new workstation was intended to reduce the number of movements above shoulder height and minimize awkward and intensive wrist movements. The workbench was designed to be at an adequate height, and the stripes and blocks are placed at a reachable distance. The new workstation does not include tasks implying arm flexion above shoulder height. However, in order to verify the suitability of this workstation and possible improvements an ergonomic assessment was performed.

*3.3. Preassembly Workers and Workstation*

The preassembly workers are all women ($n = 8$), selected by the company's supervisors considering the information of the occupational health department. This selection was based on the following criteria: prevalence of WMSD previously diagnosed affecting the upper limbs; and, workers experienced with more than 8 years in the assembly section. With the intent to collect demographic and psychophysical assessments of the preassembly tasks, the questionnaire explained in the subchapter 2.3 was applied. Concerning the demographic data, the age of these workers is 49.9 (± 7.7) years old and they have a mean of work experience of 10.9 (± 0.4) years at the assembly section. The workers were at the preassembly station for 4.8 (± 4.1) months. All of the workers reported one or more musculoskeletal problems (such as carpal tunnel syndrome, disc herniation, tendinitis).

The NMQ results (Figure 5) are related to the prevalence of musculoskeletal discomfort/pain along the last 12 months. As observed with the previous results, the McNemar test shows a perfect concordance ($p = 1.000$) between the perceptions for the last 12 months and the last 7 days across the body regions considered. These results demonstrated that the body regions with a higher incidence of musculoskeletal problems are the lumbar region and the wrists/hands. The awkward postures and repetition of actions are important risk factors for these body regions and these factors are present in the preassembly workstation. The fact that the workers still have musculoskeletal complaints indicates the need for further improvements in the workstation.

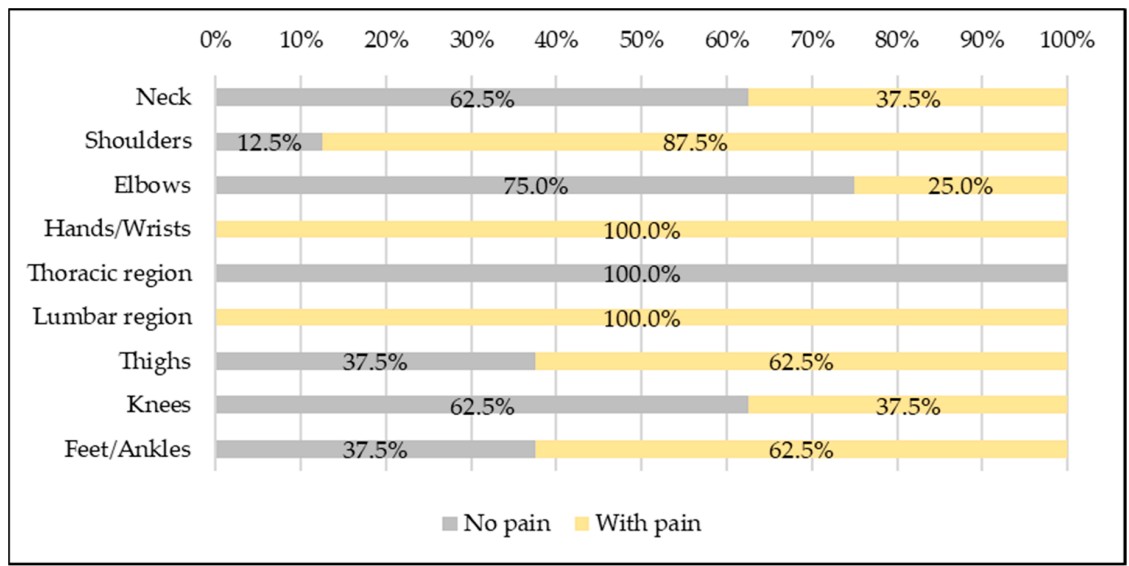

**Figure 5.** NMQ results—preassembly workers (*n* = 8).

Comparing the assembly workstation with the preassembly workstation, in general, the later obtains a better workers' evaluation which hints towards postural improvements. The global opinions about the preassembly workstation measured by the 5-Likert scale (Figure 6) demonstrated that the preassembly also improved the workers' well-being. However, some of the workers indicated that this workstation could be improved, namely: (i) the height of workbench should be lower for two of the workers; (ii) three workers suggested that the workstation should be in a less noisy place; (iii) two workers suggested the elimination of the task of gluing.

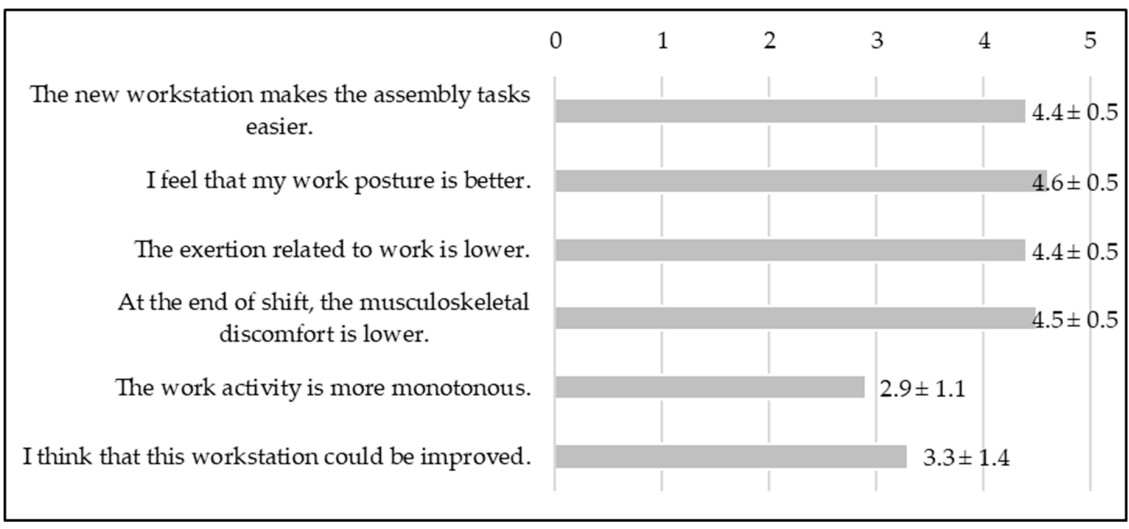

**Figure 6.** The 5-Likert scale mean results (± SD) across statements related to the quality of preassembly workstation.

Figure 7 presents the perceived exertion, assessed by CR-10 Borg scale, along with the ten preassembly tasks. The task of applying glue to the blocks presents the higher psychophysical score mean. The workers referred that the higher repetitiveness of the actions on the pistol is the most aggravating factor. They also mentioned that the task of reaching the glue pistol is more difficult when the support is unclean.

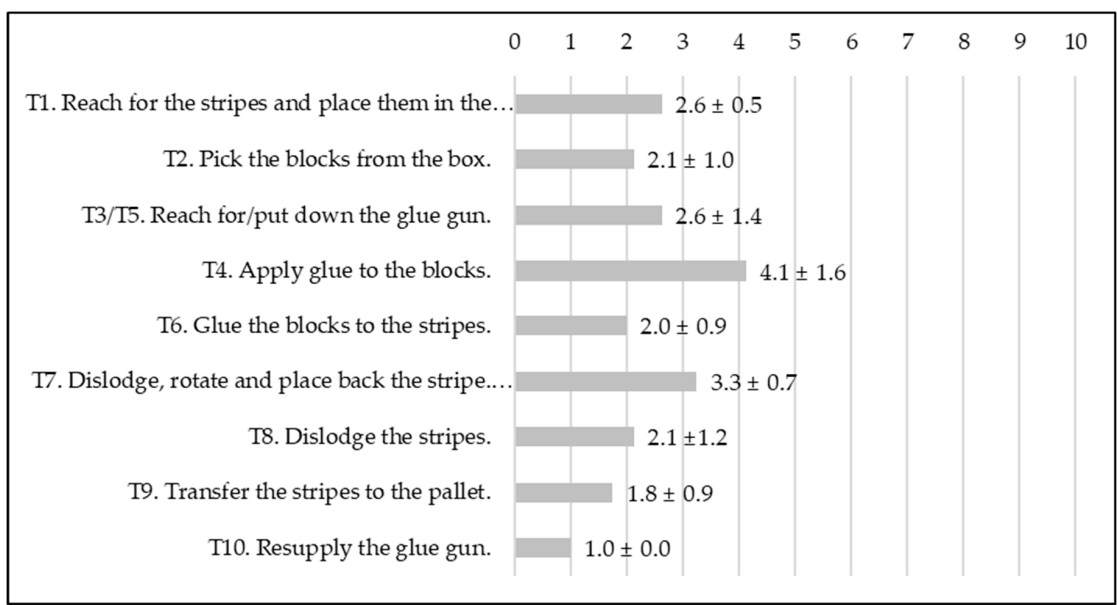

**Figure 7.** Perceived exertion (mean values ± SD) across the preassembly tasks.

Finally, the main results of the RULA and KIM-MHO assessment are presented in Table 7. The mean values indicate that Tasks 6 and 9 (Figures 8 and 9, respectively) present a higher RULA rating, indicating a higher musculoskeletal risk when compared with the other tasks. However, in Task 6 the upper limb is more affected because of the posture adopted during the gluing of the blocks to the stripes. As evidenced in Figure 8, the shape and size of the blocks handled lead to the frequent ulnar deviation and extension of the hand-wrist system. In Task 9, transferring the stripes to the pallet caused a neck extension and the flexion and inclination of the trunk, which lead to an uneven balance of the bodyweight (as shown in Figure 9). These findings demonstrated that the workstation should be redesigned for this task. Therefore, the implementation of a lifting table should be considered as well as the elimination of the lateral roller conveyor. The RULA assessment reveals relevant to define relevant corrections in the workstation. This method has been widely used during ergonomic interventions in several workstations involving repetitive tasks/movements and awkward postures [31,32].

As mentioned in the previous sub-chapter, the majority of the preassembly workers present musculoskeletal disorders affecting the wrists (carpal tunnel syndrome). This fact increases the concern about the risk assessment, which also includes a method more focused on the hand-wrist system, such as KIM-MHO. In the actual project phase and as mentioned above, the KIM-MHO was applied regarding a more complete and comprehensive ergonomic assessment. All the postures observed across the preassembly tasks were considered. However, this method allows a global postural assessment for each task, considering the most frequent postures (so the final result is not calculated as a mean value as done for RULA assessment). Klussmann et al. [33] demonstrated that KIM-MHO risk scores are significantly related to the prevalence of musculoskeletal symptoms (assessed by Nordic questionnaire) and clinical conditions especially in the shoulder, elbow and hand/wrist body regions among more than 600 employees exposed to MHO. Therefore, its application is relevant at the current study, since the tasks assessed are MHO and the selected workers have musculoskeletal problems at hand/wrist.

**Table 7.** Summary of Rapid Upper Limb Assessment (RULA) and Key Indicator Method (KIM) assessment (bold denotes the major value for each assessment).

| | RULA | | KIM-MHO | |
| --- | --- | --- | --- | --- |
| **Task** | **Rating Mean (SD)** | **Risk Level** | **Risk Score** | **Risk Level** |
| Task 1—Reach a stripe and align. | 3.2 (0.4) | **II** | 48 | **III** |
| Task 2—Reach blocks and stack. | 3.6 (0.9) | **II** | 48 | **III** |
| Task 3—Reach the glue pistol. | 3.0 (0.0) | **II** | 34 | **III** |
| Task 4—Apply glue to the blocks. | 3.0 (0.0) | **II** | 64 | **IV** |
| Task 5—Put the glue pistol on the support. | 3.0 (0.0) | **II** | 34 | **III** |
| Task 6—Fix blocks on the stripe. | **4.4** (0.5) | **II** | 48 | **III** |
| Task 7—Relocate or reverse the stripe. | 3.0 (0.0) | **II** | 30 | **III** |
| Task 8—Take off stripe. | 3.2 (0.4) | **II** | 40 | **III** |
| Task 9—Transfer stripe to the pallet. | **4.4** (1.3) | **II** | 46 | **III** |
| Task 10—Supply the glue pistol. | 3.0 (0.0) | **II** | 44 | **III** |

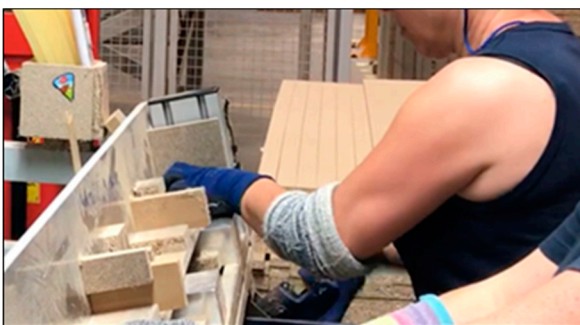

**Figure 8.** Posture during Task 6.

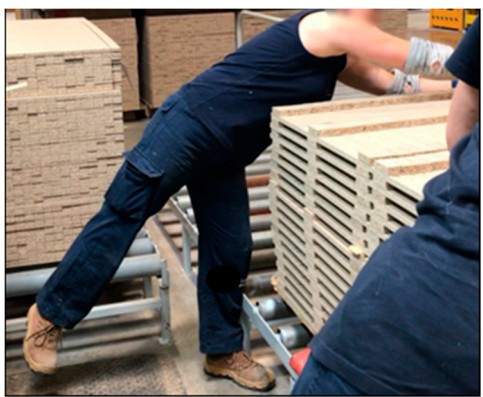

**Figure 9.** Posture during Task 9.

The KIM-MHO findings pointed out to the fact that Task 4 (apply glue to the blocks—Figure 10) is associated with a higher physical workload (high load situation), indicating that the redesign of the workplace is necessary to prevent WULD. However, all the tasks present a significant risk, being the duration/time rating the main contributor. According to the KIM-MHO, this risk factor is equivalent to the total duration of the activity per shift, calculated by the cycle duration and number of repetitions per shift. The value obtained for this factor is transversal to all preassembly tasks because these are parts of the same work cycle. Therefore, considering that the selected workers have a history of WULD and several musculoskeletal complaints (evidenced by the questionnaire results), the redesign of the preassembly workstation must be developed in the future project phases.

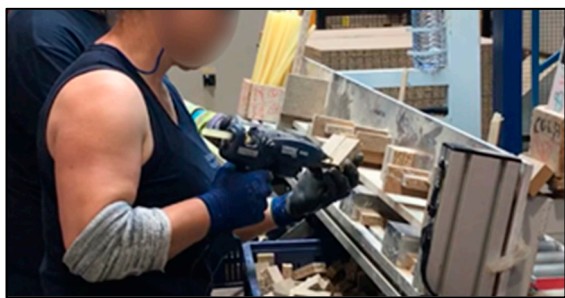

**Figure 10.** Posture during Task 4.

*3.4. Definition of the Ergonomic Requirements and Future work*

The ergonomic study presented was developed regarding the future implementation of a COBOT to support the assembly tasks described. Even taking into account ergonomic criteria, this workstation still has limitations that we intended to mitigate with the COBOT. In order to design and implement workstations with robots, one needs to consider the worker as the most flexible and fragile component of the equation; therefore, the study of work conditions and tasks is essential [34]. The current study exemplifies this and presents a guide to develop an ergonomic study that could support the design of an assembly task with a COBOT.

Summarily, the FIOH method allows the characterization of the initial problem, identifying/screening the main risk-factors that compromise the wellbeing and health of the workers in an industrial setting. This method is essential to support the design of a new workstation. Additionally, the collection of the workers' perceptions about the workstation and musculoskeletal symptoms is important to ensure the workers' involvement and to support the identification of problems. This approach consists of a participatory ergonomics intervention, incorporating the workers along with the study and foreseeing the success of interventions at the workstations [35]. Based on this assumption, the workers' opinions must be included in future evaluations.

Moreover, we also applied and recommended the application of observational methods more specific for ergonomic assessment of the tasks. In this case, the assembly tasks present different WMSD risk factors and the workers who performed these tasks suffer from musculoskeletal problems (as evidenced by the questionnaires' results). Khan et al. [36] emphasize that WMSD amongst assembly workers has registered a steeper increasing trend when compared to other industrial activities, with significant associated productivity drops. This problem is often associated with physical risks at the workstations (e.g., repetitive handling tasks, awkward postures) [3]. Therefore, RULA and KIM-MHO methods are adequate to assess these risks and support the future task allocation for the hybrid human-robot team.

Globally, the results of the ergonomic assessment (Table 7) indicate that robotics implementation should eliminate Task 4 in order to reduce WULD risk. The workers also pointed to this task as the most physically demanding in the preassembly process (as evidenced by the scores represented in Figure 7). However, for the redesign of the preassembly workstation, anthropometric data will be necessary to correct work postures [37] related to different preassembly tasks (mainly, the tasks 6 and 9). This redesign should improve the physical organization of the entire workstation, including the palletization area.

The current study allows the definition of a set of requirements for the future development of a workstation with collaborative robotics. As mentioned previously, this definition of requirements was tripartite, including workers, researchers and company's practitioners. However, Fletcher et al. [38] state that the current problem is the inexistence of a standard valid framework and further research is required to define a design framework for the future human-robot assembly systems. Figure 11 summarizes this ergonomic study and could be a guide to future works focused on the design of assembly workstation with COBOT, integrating ergonomic requirements.

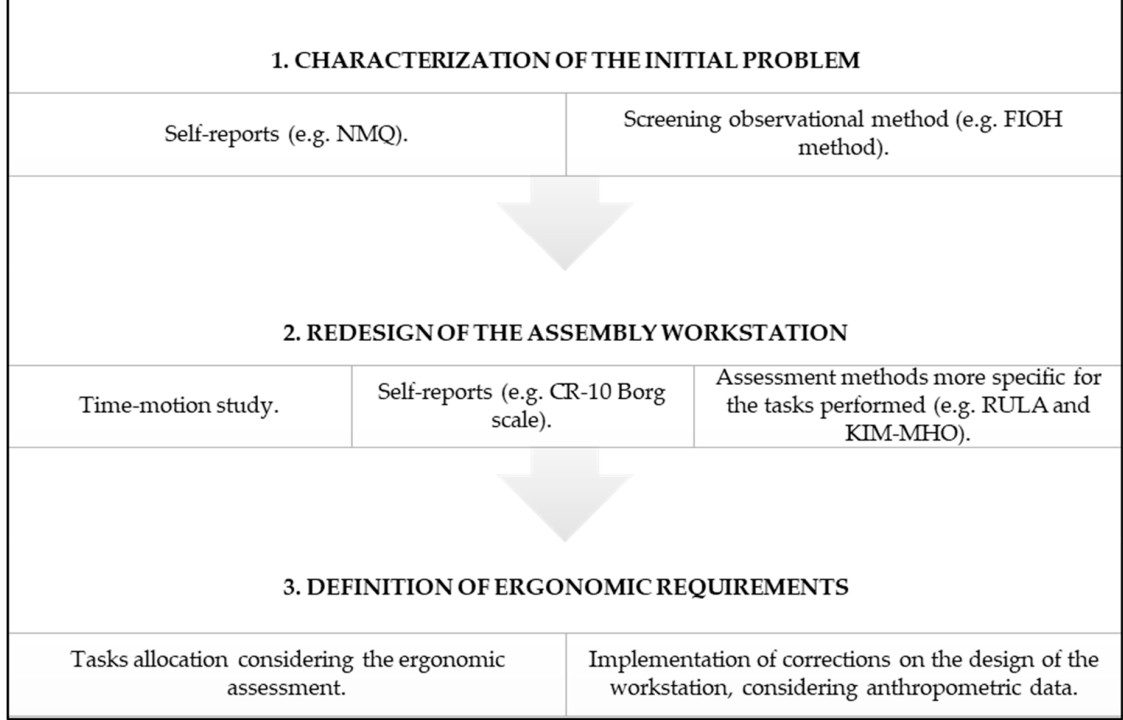

**Figure 11.** Proposal of a simplified framework for the design of assembly workstations with collaborative robotics.

This comprehensive analysis was conducted in line with the recommendations presented by Battini et al. [5]. These authors propose a theoretical framework to assembly systems design taking into account technological variables (such as work times and methods), environmental variables (e.g., absenteeism, staff turnover), and ergonomic assessments to obtain a comprehensive analysis. Based on this study the main ergonomic requirements for the future workstation are listed below:

(i) Replace/eliminate the task of apply glue, since it is a critical task according to ergonomics' point of view and physically demanding due to the actions repeated. Moreover, the burns motivated by the hot glue constitute the accident more frequent in this section;

(ii) Height-correction of the workbench to accommodate the workers' variability in terms of anthropometric data [37], located between 1066 mm (95th percentile of shoulders height of the Portuguese adult women) and 914 mm (5th percentile of shoulders height of the Portuguese adult women);

(iii) Diversification of the work content through the inclusion of different tasks/interactions between workers and/or COBOT (the preassembly workstation is monotonous and repetitive);

(iv) Implementation of corrective measures to reduce noise exposure, which difficult communication and increases muscular tension;

(v) Correct the baseboard location, eliminating the lateral conveyor or introducing a rotative table for the palletizing zone.

Regarding the productivity requirements, it is expected that new interactions of the preassembly process should match the cycle times calculated with the current process, refer to Table 6. Any reduction of cycle times and consecutive productivity improvement will be regarded as a positive outcome. However, such an upgrade shall not compromise the ergonomics requirements, which is the project's primary objective.

In fact, manual assembly work presents high flexibility, but low productivity compared to a fully automated assembly system. The implementation of the COBOT in this type of workstation could improve productivity maintaining flexibility. In the future assembly system, human workers provide

manual work and the design of the system should be oriented by an adaptive utilization of human capabilities, foreseeing the improvement of productivity and workers' wellbeing [38]. The hybrid team composed by humans and robot will support the demographic diversity of workers, as well as their physical limitations, where robots help or take over the most demanding physical tasks.

**Author Contributions:** A.C., conceptualization, methodology; A.C. and C.F., investigation and writing; A.C.B., satistical analysis; N.S., investigation support; L.R., project supervision; P.C., N.C. and P.A., writing review and validation. All authors have read and agreed to the published version of the manuscript.

**Funding:** This work has been supported by NORTE-06-3559-FSE-000018, integrated in the invitation NORTE-59-2018-41, aiming the Hiring of Highly Qualified Human Resources, co-financed by the Regional Operational Programme of the North 2020, thematic area of Competitiveness and Employment, through the European Social Fund (ESF). This work has been also supported by FCT – Fundação para a Ciência e Tecnologia within the R&D Units Project Scope: UIDB/00319/2020.

**Conflicts of Interest:** The authors declare no conflict of interest.

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
