# Peer review of "Towards an Ergonomic Assessment Framework for Industrial Assembly Workstations—A Case Study"

_applsci, doi:10.3390/app10093048_

Round 1

Reviewer 1 Report

the paper needs a few bits of English revisions( see lines 38, 44 and figure 6-more easy should be easier) .

Tables/ figures should be able to fully stand on their own-Figure 3 needs clarification of numbers on far left, table 6 needs unit of time.

Discussion should discuss some limitations a bit better- for example- was there concern that the female participants of the study were selected by the company- what were criteria, were they representative, and how generalizable.

Author Response

My co-authors and I would like to thank the anonymous reviewers for their valuable review of our work. The comments and suggestions of the reviewers were very useful to hopefully improve the quality of the manuscript. We have generally accepted all the reviewers’ suggestions/comments and we include below all of our responses/comments on the points addressed.

Reviewer #1

  1. Comment: “the paper needs a few bits of English revisions ( see lines 38, 44 and figure 6-more easy should be easier).

RESPONSE: Thank you for the comment; The paper was entirely revised, including the lines 38, 44 and Figure 6.

  1. Comment: “Tables/ figures should be able to fully stand on their own-Figure 3 needs clarification of numbers on far left, table 6 needs unit of time.”

RESPONSE: It was revised as requested (in Figure 3 and Table 6).

  1. Comment: “Discussion should discuss some limitations a bit better- for example- was there concern that the female participants of the study were selected by the company- what were criteria, were they representative, and how generalizable.”

RESPONSE: The current study corresponds to a case study as stated in the manuscript, so it was not intended for data generalization. However, we explained that the assembly section is composed of 14 similar workstations with 60 female workers. A new preassembly workstation was designed for workers with musculoskeletal problems/complaints and/or medical restrictions to perform certain tasks. Therefore, and as mentioned in section 3.3: “preassembly workers are all women (n = 8), selected by the company’s supervisors considering the information of the occupational health department. This selection was based on the following criteria: prevalence of WMSD previously diagnosed affecting the upper limbs; and, workers experienced with more than 8 years in the assembly section.” This information was detailed in the revised manuscript (Lines 340-343).

Reviewer 2 Report

The work is well written and contextualizes a case study in an important domain. In my opinion, small adjustments and explanations in writing can improve the work. Suggestions follow below.

- Line 150 – Figure 1 shows one example of the workstation assembly. In line 118 authors have “14 similar workstations”. Then, as a suggestion, the legend of figure 1 must include “example of the existing assembly workstation”.

- Line 282 - “ …. The NMQ results presented in Figure 4 ….” Figure 4 must be replaced by Figure 2.

Concerning Figures 2, 3, 5, 6, 7 and 11

- The figures 2, 3, 5, 6, 7 and 11 can be improved, namely with a better description and legend. As an example, in figure 5, what is the meaning of Yes/No ? In figure 3, the “topic assessed” must be included as a text description and not with a topic number. Text description in figure 11 can be visually improved.

Concerning Table 6

- In table 6, it should be explained how the average times and standard deviation were obtained and calculated.

- Concerning volunteer workers

The authors refer to the existence of 60 females as workforce. However, they select a sample of 44 volunteers.

It is not clear, and must be well explained, why 44 volunteers were chosen and what were the criteria for choosing these and eliminating the others.

In addition, among the group of 44 volunteers, 17 described medically diagnosed musculoskeletal problems. However, they were all considered to be in the same group. Why the possibility of separating this sample into a group of 27 healthy volunteers and 17 volunteers with problems was not considered?

- Concerning the Statistical Analysis:

Line 253 – 2.4. Statistical Analysis

- Authors can explain more clearly the reason for choosing the Mcneymar and Friedman tests.

Author Response

My co-authors and I would like to thank the anonymous reviewers for their valuable review of our work. The comments and suggestions of the reviewers were very useful to hopefully improve the quality of the manuscript. We have generally accepted all the reviewers’ suggestions/comments and we include below all of our responses/comments on the points addressed.

Reviewer #2

1. Comment: “Line 150 – Figure 1 shows one example of the workstation assembly. In line 118 authors have “14 similar workstations”. Then, as a suggestion, the legend of figure 1 must include “example of the existing assembly workstation”.”

RESPONSE: It was revised according to reviewer suggestion (Line 150 – Figure 1).

2. Comment: “Line 282 - “ …. The NMQ results presented in Figure 4 ….” Figure 4 must be replaced by Figure 2”.

RESPONSE: Figures were replaced as suggested (Line 286 in the revised manuscript).

3. Comment: “Concerning Figures 2, 3, 5, 6, 7 and 11 - The figures 2, 3, 5, 6, 7 and 11 can be improved, namely with a better description and legend. As an example, in figure 5, what is the meaning of Yes/No ? In figure 3, the “topic assessed” must be included as a text description and not with a topic number. Text description in figure 11 can be visually improved.”

RESPONSE: Figures’ legends were improved/revised as suggested.

4. Comment: “In table 6, it should be explained how the average times and standard deviation were obtained and calculated.”

RESPONSE: The average times and SD were calculated by a time study, observing different work cycles. Information about these results was added in Lines 323 and 324.

5. Comment: “Concerning volunteer workers, the authors refer to the existence of 60 females as workforce. However, they select a sample of 44 volunteers. It is not clear, and must be well explained, why 44 volunteers were chosen and what were the criteria for choosing these and eliminating the others.”

RESPONSE: The assembly section has 3 work shifts, with a total of 60 workers. However, we had 44 volunteers to participate. We didn’t exclude any worker but we have only considered those who volunteered. Therefore, we added a note about this in Line 278.

6. Comment: “In addition, among the group of 44 volunteers, 17 described medically diagnosed musculoskeletal problems. However, they were all considered to be in the same group. Why the possibility of separating this sample into a group of 27 healthy volunteers and 17 volunteers with problems was not considered?

RESPONSE: We acknowledge the reviewer’s comment. We consider that this could be an interesting topic for future research, as well as other individual factors such as the workers’ aging. We didn’t do this in the discussion sector because it was not the objective of the current study.

In addition, we added more information about these assembly workers (Lines 288 and 289 ): “The results demonstrate that all of the inquired workers reported any musculoskeletal symptoms, even the workers without previously disorders diagnosed.” And, as referred in the manuscript, a new preassembly workstation was designed for workers with musculoskeletal problems/complaints and/or medical restrictions to perform certain tasks. Therefore, and as mentioned in 3.3 section: “preassembly workers are all women (n = 8), selected by the company’s supervisors considering the information of the occupational health department. This selection was based on the following criteria: prevalence of WMSD previously diagnosed affecting the upper limbs; and, workers experienced with more than 8 years in the assembly section.” This information was detailed in the revised manuscript (Lines 340-343).

7. Comment: “Line 253 – 2.4. Statistical Analysis - Authors can explain more clearly the reason for choosing the Mcneymar and Friedman tests.”

RESPONSE: We revised section 2.4 to include the more clear reason for our test selection and we marked the changes in the revised manuscript.